# The Central Fluid Percussion Brain Injury in a Gyrencephalic Pig Brain: Scalable Diffuse Injury and Tissue Viability for Glial Cell Immunolabeling following Long-Term Refrigerated Storage

**DOI:** 10.3390/biomedicines11061682

**Published:** 2023-06-10

**Authors:** Mark Pavlichenko, Audrey D. Lafrenaye

**Affiliations:** 1Department of Anatomy and Neurobiology, Virginia Commonwealth University, Richmond, VA 23298-0709, USA; 2Richmond Veterans Affairs Medical Center, Richmond, VA 23249-4915, USA

**Keywords:** traumatic brain injury, axonal injury, micro pig, diffuse pathology, microglia, aged tissue

## Abstract

Traumatic brain injury (TBI) affects millions of people annually; however, our knowledge of the diffuse pathologies associated with TBI is limited. As diffuse pathologies, including axonal injury and neuroinflammatory changes, are difficult to visualize in the clinical population, animal models are used. In the current study, we used the central fluid percussion injury (CFPI) model in a micro pig to study the potential scalability of these diffuse pathologies in a gyrencephalic brain of a species with inflammatory systems very similar to humans. We found that both axonal injury and microglia activation within the thalamus and corpus callosum are positively correlated with the weight-normalized pressure pulse, while subtle changes in blood gas and mean arterial blood pressure are not. We also found that the majority of tissue generated up to 10 years previously is viable for immunofluorescent labeling after long-term refrigeration storage. This study indicates that a micro pig CFPI model could allow for specific investigations of various degrees of diffuse pathological burdens following TBI.

## 1. Introduction

Traumatic brain injury (TBI) is a major health care concern that carries significant personal and societal burdens [1,2,3,4]. There are approximately 2 million reported cases of TBI annually in the United States, with global cases reaching 50 million each year, with the understanding that the real numbers are likely higher due to the vast majority of TBIs being mild and going unreported [4,5,6,7]. Although our knowledge of the complex pathologies associated with TBI has progressed, understanding of diffuse pathologies, which are associated with morbidity in the human population [8,9,10], is still limited. 

Diffuse traumatic axonal injury (TAI) is one of the pathological hallmarks of TBI, and is commonly used in experimental mild TBI studies to convey the degree of injury [11,12,13,14,15,16,17,18]. Neuroinflammation has become another key pathological feature of brain injury in both human and animal studies [8,19,20,21]. Many recent studies have demonstrated the impact of inflammatory cascades in regulating behavioral morbidities, general pathology, and neuronal function in both the normal brain and in various disease states, including TBI [22,23,24,25]. Studies have demonstrated neuroinflammation in the human population following TBI in various brain regions, with consistent involvement of the thalamic domain [8,19,20,21]. Microglia, the innate immune cells of the brain, are critical mediators of these TBI-induced neuroinflammatory processes [26,27,28,29,30,31,32,33]. Astrocytes are key regulators of various processes, including maintenance of the blood–brain barrier, water movement, synaptic activity regulation, glucose storage, and neuroinflammatory processes [30,34,35,36]. Serum levels of glial fibrillary acidic protein (GFAP), a component of the astrocyte cytoskeleton, are also correlated with negative outcomes following TBI and is one of two currently FDA-approved biomarkers for TBI [37,38,39].

We currently do not have the technology to specifically investigate TBI-induced diffuse pathological progression at an individual cellular level in the living human brain; therefore, animal models are used to tease out potential targets for further study. Many therapeutics have shown promising results in rodent models of TBI, unfortunately, this efficacy has been limited when translated to humans [40,41,42]. The high reliance on lower-order species for preclinical TBI research has been implicated in this translational failure, in that some processes and mechanisms found in rodents do not occur in humans, and vice versa [41,43]. Therefore, there have recently been calls for more models using higher order mammals with inflammatory responses and brain cytoarchitecture more similar to humans, such as pigs, to better evaluate potential therapeutics prior to human translation [42,44,45,46,47,48,49]. 

With this goal in mind, we began characterization of a central fluid percussion injury (CFPI) model of mild diffuse TBI in the adult micro pig [50,51,52,53]. We found that CFPI resulted in diffusely dispersed injured axons and activated microglia in various brain regions without producing focal lesions or gross tissue damage [51]. We further found that this model recapitulated diffuse pathological and serum biomarker profiles of the human population [50,51,52]. As human injury occurs on a spectrum, the current investigation aimed to evaluate the potential for graded levels of injury-induced pathology in the micro pig model of CFPI. Additionally, due to the expense of generating large animal TBI models, we sought to determine the duration that our previously generated micro pig brain tissue would maintain viability for immunolabeling. 

## 2. Materials and Methods

### 2.1. Animals

Data from animals used in this investigation have been published previously [51,52]. Tissue from one recently generated animal was used as freshly harvested control tissue for our immunolabeling studies. We used tissue from a total of twenty-two ~6-month-old adult male Yucatan micro pigs (*n* = 3 sham; *n* = 18 injured, *n* = 1 new tissue), weighing 15–25 kg at time of injury. Prior to injury induction, animals were housed in pairs on a 12 h light–dark cycle in pens with free access to food and water. Pens had epoxy flooring with a drain in the back of the pen and a mixture of solid cinderblock wall and fencing. Following injury, pigs were single housed in a separate pen within the same room. All experimental protocols were approved by the Virginia Commonwealth University and Richmond Veterans Affairs Medical Center Institutional Animal Care and Use Committees, AAALAC international-accredited and USDA-registered organizations. The Virginia Commonwealth University and Richmond Veterans Affairs Medical Center Institutional Animal Care and Use Committees adhere to all regulations outlined in the “Guide for the Care and Use of Laboratory Animals: 8th Edition” (National Research Council [54]). 

### 2.2. Surgical Preparation and Injury Induction

As previously published [51], micro pigs were initially sedated with an intramuscular injection in the rear flank of 100 mg/mL Xylazine (2.2 mg/kg; AnaSed Injection, Shenondoah, IA, USA), 100 mg/mL Telazol (2.0 mg/kg; Tiletamine HCL and Zolazepam HCL; Pfizer, New York, NY, USA), and 0.01 mg/kg Glycopyrrolate. After initial sedatives took full effect (5–10 min post injection), sodium pentobarbital (60 mg/kg; Sigma-Aldrich, St. Louis, MO, USA) was administered intravenously through a superficial ear vein. Loss of the palpebral reflex, in which the canthus of the palpebral fissure of the eye was touched gently to elicit a reflexive eye twitch/closing, and jaw tone were used to verify the plane of anesthesia. Once 2–3 gentle touches to the canthus of the palpebral fissure did not elicit a response, and the jaw was easily manipulated, the micro pig was intubated with a size 6–7 endotracheal tube and ventilated with 1–2% isoflurane mixed in 100% oxygen throughout the experiment. Ophthalmic lubricant was applied to avoid damage or drying of the eye during surgery or recovery. Body temperature was monitored with a rectal thermometer and maintained at 37 °C by adjusting the level of the self-heating surgical table. Chux underpads and/or blankets were placed between the stainless-steel surface of the surgical table and the pig to reduce chances of burns. The head and inner thigh were shaved with electric clippers and sanitized with betadine and 70% ethanol wipes. Heparin-coated catheters were inserted within the right femoral artery and vein and sutured into place for continuous monitoring of mean arterial blood pressure (MABP; via the arterial line), assessment of blood gases via the arterial line, and replacement of fluids with Lactated Ringer’s (Hospira, Lake Forest, IL, USA) via the venous line to maintain hydration. Following the canula placement, the animal was placed on its sternum and the animal’s head was shaved, sterilized with betadine and 70% ethanol wipes, and draped. A midline incision was made from the supraorbital process to the nuchal crest and a 16 mm diameter circular craniotomy was trephined along the sagittal suture with a manual surgical trephine (Medicon 57-63-16), positioning the center of the craniotomy 14 mm anterior to lambda (on the nuchal crest) and leaving the Dura intact (Figure 1A). A custom stainless-steel threaded hub (Custom Design and Fabrication, Richmond, VA, USA) with an outer diameter of 17 mm and an inner diameter of 14 mm (Figure 1B) was screwed into the craniotomy site to a depth of ~4 mm. Dental acrylic (methyl-methacrylate; Hygenic Corp., Akron, OH) was applied around the hub to insure hub stability. The hub was filled with sterile saline and wet gauze was placed over the dental acrylic to serve as a heat sink for the exothermic reaction that occurs during hardening. 

The induction of the central fluid percussion injury (cFPI) was carried out as described previously [50,51]. Following verification that the dental acrylic was fully hardened, anesthetized micro pigs were connected to the cFPI device that had been fitted with an L-shaped stainless steel adaptor onto which the injury hub locked (Figure 1B,C). This adaptor allowed for the redirection of the fluid pulse downward through the injury hub on the ventral surface of the pig’s skull. Following connection and verification that bubbles hadn’t been introduced to the closed fluid-filled system, micro pigs were injured with a fluid pressure pulse at a magnitude of 1.39–1.83 atmospheres and duration of 24–32 msec (Appendix A) measured by a transducer affixed to the injury device and displayed on an oscilloscope (Tektronix, Beaverton, OR, USA). Immediately after injury induction, animals were disconnected from the injury device, the set screws were unscrewed from the skull, and the dental acrylic, hub, and screws were removed as a unit. The hub was then removed from the surrounding dental acrylic for sterilization and reuse. While there was some sub-arachnoid bleeding following injury, the cFPI did not result in any breach of the Dura mater or induce subdural hematoma formation. Gel foam was placed over the craniotomy/injury site to alleviate small amounts of bone bleeding, and the scalp was sutured without replacing the bone to the craniotomy site. 

### 2.3. Physiology

Systemic physiology was monitored and recorded throughout the duration of anesthesia, both prior to cFPI and throughout the post-injury monitoring period. Mean arterial blood pressure was measured through the canulated femoral artery, hemoglobin oxygen saturation was monitored via a pulse oximeter, and a rectal thermometer was used to measure body temperature. All physiological readings were measured and visualized by a Cardell^®^ MAX-12HD (Sharn veterinary, Inc., Chicago, IL, USA) system. The partial pressures of oxygen and carbon dioxide in arterial blood, PaO_2_, and PaCO_2_, respectively, hematocrit (Hct), bicarbonate (HCO_3_), hemoglobin (Hb), and pH values were assessed on blood draws from the femoral arterial cannula using a Stat Profile pHOx blood gas machine (NOVA Biomedical, Waltham, MA, USA). The summarized physiological results for these animals have previously been published [51]. However, this manuscript investigates individual animals’ pre- and post-injury physiological readouts (Appendix A).

### 2.4. Tissue Processing

At terminal endpoints of 6 h post-cFPI, micro pigs were overdosed with 3 mL euthasol euthanasia-III solution (Henry Schein, Dublin, OH, USA) and immediately transcardially perfused with 6 L of 0.9% saline followed by 8 L of 4% paraformaldehyde/0.2% gluteraldehyde in Millonig’s buffer (136 mM sodium phosphate monobasic/109 mM sodium hydroxide), as previously published [51]. After transcardial perfusion, the brains were removed from the skull and post-fixed in 4% paraformaldehyde/0.2% gluteraldehyde/Millonig’s fixation buffer for an additional 48–72 h at 4 °C. Postfixed brains were blocked into 5 mm coronal segments throughout the rostral–caudal extent using a small pig brain slicer matrix (Cat. # PBMPBS050-1 Zivic Instruments, Pittsburgh, PA, USA). Segments containing the thalamus or corpus callosum were bisected at the midline, and the left side was further processed. All segmented brain tissue not used in active studies was cataloged, cryoprotected in 30% sucrose, and frozen in tissue tek Optimal Cutting Temperature (O.C.T.) compound (Tissue-Tek #4583; Sakura Finetek; Torrance, CA, USA) prior to freezing and storage at −80 °C for future use. The 5 mm coronal segments containing the left hemi-thalamus or corpus callosum were never frozen, rather, they were embedded in agarose and coronally sectioned in 0.1 M phosphate buffer with a vibratome (Leica, Banockburn, IL, USA) at a thickness of 40 µm. Sections were collected serially in 6-well plates (240 μm between sections in each well) and stored in Millonig’s buffer at 4 °C. Thalamic tissue treated in this manner was stored in the refrigerator for multiple years prior to investigations of viability for immunobiological analysis. Buffer was changed and/or added every few years to avoid dehydration. 

### 2.5. Immunohistochemistry

#### 2.5.1. Previous Labeling of Thalamic and Corpus Callosal Tissue for Axonal Injury and Microglial Activity Index Assessments

All immunohistological labeling and analyses used to determine injury model scalability was carried out on freshly generated tissue during the studies published previously [51]. Mean thalamic data for injured and sham groups were previously published in Lafrenaye et al., 2015 [51]; however, corpus callosum data were not reported. For immunohistological labeling, a random well (1–6) was selected using a random number generator and six pieces of tissue containing the thalamic or twelve sections containing the corpus callosum were taken for immunolabeling. Tissue from all animals was processed concomitantly to reduce variability between animals. 

To visualize injured axons, fluorescent immunohistochemistry against amyloid precursor protein (APP) was used to detect axonal transport issues indicative of axonal injury [13,14,55]. In this procedure, sections were blocked and permeabilized at room temperature in 5% normal goat serum (NGS), 2% bovine serum albumin (BSA), and 1.5% triton in phosphate-buffered saline for 2 h followed by overnight incubation with a rabbit antibody against the C-terminus of β-APP (1:700; Cat.# 51-2700, Life Technologies, Carlsbad, CA, USA) at 4 °C in 5% NGS/2% BSA. Tissue was washed with 1%NGS/1%BSA in PBS at least six times prior to secondary antibody incubation with Alexa Fluor 568-conjugated goat anti-rabbit IgG (1:700; Cat.# A-11011, Life Technologies, Carlsbad, CA, USA) in 1%NGS/1%BSA/PBS at room temperature for 1 h. Tissue was washed in 1%NGS/1%BSA in PBS at least six times. Tissue was mounted on slides using Vectashield hardset mounting medium with Dapi (Cat.#H-1500; Vector Laboratories, Burlingame, CA, USA).

For the visualization of microglia, chromatic immunohistochemistry against ionized calcium-binding adaptor molecule 1 (Iba-1) was carried out. Endogenous peroxidases were quenched with hydrogen peroxide prior to chromatic immunolabeling. Tissue was blocked and permeabilized in 1.5% triton/10% NGS/PBS for 2 h followed by incubation with a rabbit antibody against Iba-1 (1:1000; Cat.#019-19741 Wako; Richmond, VA, USA) in 10% NGS/PBS at 4 °C. Tissue was washed with 1%NGS/1%BSA in PBS at least six times prior to secondary antibody incubation with biotinylated goat anti-rabbit IgG (1:1000; Cat.# BA-1000, Vector Laboratories, Burlingame, CA, USA). To enhance the chromatic signal, tissue was incubated in avidin biotinylated enzyme complex using the Vectastain ABC kit (Vector Laboratories, Burlingame, CA, USA). The substrate used for the chromatic reaction to visualize microglia was 0.05% diaminobenzidene/0.01% H_2_O_2_/0.3% imidazole/PBS. Following cessation of the reaction, tissue was mounted, dehydrated in a series of alcohols, and cover-slipped with Permount mounting media (Fisher Scientific, Cat#SP15).

#### 2.5.2. Labeling of Stored Thalamic Tissue for Immunohistochemistry Efficacy Assessments

Archived thalamic tissue from 19 different animals (generated between 25 July 2013 and 21 July 2015) that had been perfusion-fixed with 4% paraformaldehyde/0.2% glutaraldehyde followed by at least 3 days of post-fixation and stored at 4 °C were used for immunohistological labeling. Data from these animals have been published previously [51]. Fixed tissue used in this study was sectioned in 0.1 M dibasic sodium phosphate buffer within 3 months of being generated and stored long term at 4 °C in Millonigs buffer (136 mmol/L sodium phosphate monobasic/109 mmol/L sodium hydroxide). Glial fibrillary acidic protein (GFAP) was used to label astrocytic soma and processes, and Iba-1 was used to label microglial soma and processes. Tissue was sequentially co-labeled for both markers, with the majority of samples (*n* = 13) being initially labeled for GFAP followed by Iba-1, and the remaining samples (*n* = 6) being labeled for Iba-1 first followed by GFAP labeling. For this, tissue was washed in PBS 3 times for 5 min, blocked, and permeabilized for 1 h at room temperature in 5% NGS/2% BSA/1.5% Triton/PBS, and incubated overnight at 4 °C with either mouse anti-GFAP primary antibody (1:1000; Cat #MAB360; Millipore) or rabbit anti-Iba-1 primary antibody (1:1000; Waco). Tissue was then washed 6 times for 5 min in 1% NGS/1% BSA/PBS prior to 1 h incubation at room temperature with either goat anti-mouse Alexa 568 (1:500, Cat#;) for GFAP visualization, or goat anti-Rabbit Alexa 488 (1:500, Cat#;) for Iba-1 visualization. Tissue was washed again 4 times for 5 min in PBS. The single-labeled tissue was then blocked again for 1 h at room temperature in 5% NGS/2% BSA/PBS and immunolabeled with the other primary antibody, either rabbit anti-Iba-1 or mouse anti-GFAP, followed by the secondary antibody, either goat anti-rabbit Alexa 488 for Iba-1 visualization or goat anti-mouse Alexa 568 for GFAP visualization, as above. Tissue was mounted with Vectashield Hardset mounting medium with DAPI prior to imaging. 

### 2.6. Image Analysis 

The imaging of APP and microglia used for the correlation matrices was carried out prior to 2015 using a Nikon Eclipse 800 microscope (Nikon, Tokyo, Japan) equipped with an Olympus DP71 camera (Olympus, Center Valley, PA, USA). Image acquisition settings were held consistent for all animals and the regions of interest were restricted to the thalamus and corpus callosum using anatomical landmarks. Imaging was carried out by an investigator blinded to animal identity.

#### 2.6.1. Quantitative Image Analysis of Diffuse Axonal Injury 

For the APP assessments in the thalamus, a total of 60 images (10 images in each of the 6 sections assessed) were taken at 10× magnification (0.72 mm^2^ field) in a systematically random fashion starting at the dorsal lateral aspect of the thalamus. For the corpus callosum, a total of 24 images (2 images in each of 12 sections) were taken at 10× magnification, which covered the majority of the region of interest. A nuclear DAPI label was used for field advancement and to verify focus as well as restriction within the region of interest. Thresholded masks of APP intensity were generated in ImageJ software (NIH, Bethesda, MD). The particle analysis tool in ImageJ was then used to assess the number of APP^+^ axonal swellings in each image (particle analysis settings circularity = 0.2–1, size = 20-infinity). The number of APP^+^ swellings per unit area was quantified for each image and averaged for each animal. 

#### 2.6.2. Quantitative Image Analysis of Microglial Activity Index 

Assessment of microglia activation for the thalamus was published previously [51]. For this the entire thalamus or corpus callosum was assessed for each of the sections selected for each animal (6 sections for the thalamus and 12 sections for the corpus callosum). Identification of microglia activation was based on specific morphological criteria. Microglia with highly ramified fine process networks that were lightly labeled with Iba-1 were considered non-reactive, while microglia with thicker, shorter, or absent processes and darker Iba-1 labeling were identified as active/reactive [55,56,57,58]. The degree of microglia activation was assessed using an index from 0 to 5 in which 0 = no microglial activation observed, 1 = ramified microglia with thicker processes and darker Iba-1 labeling observed in ~5% of the region of interest, 2 = activated microglia observed in ~5–10% of the region of interest, 3 = activated microglia observed in ~10 < 25% of the region of interest, 4 = activated microglia observed in ~25 < 50% of the region of interest and 5 = activated microglia observed in >50% of the region of interest. Two blinded investigators analyzed all sections independently, and their scores were averaged for each animal. 

#### 2.6.3. Qualitative Analysis of Glial Labeling in Old Pig Tissue

Immunofluorescent labeling of microglia and astrocytes in pig tissue generated >7 years prior and stored at 4 °C was qualitatively assessed by investigators blind to the age of tissue. Images were taken with a Keyence BZ-X800 microscope (Keyence Corporation of America, Itasca, IL, USA) at 40× magnification with all settings held constant for each sample. Experimenters randomly selected 6 areas of each section within the thalamic domain to evaluate the degree of labeling of Iba-1+ microglia and GFAP+ astrocytes. A high-degree label was associated with widely distributed and clearly visible cell soma and processes, a medium-degree label was associated with sporadically distributed and visible cell soma and/or processes, and a low/no-degree label was associated with no visible cell soma or processes. Age of tissue was assessed for each labeling category (high, medium, and low/no) to gain insight regarding the potential viability of tissue for immunohistochemistry over time. 

### 2.7. Statistical Analysis

Two-tailed unpaired T-tests with equal variance not assumed were performed to assess corpus callosal axonal injury and microglial activation index between sham and cFPI animals. Individual animal correlation data were analyzed with a Spearman’s Rho Correlation analysis due to its lack of reliance on the assumption of data normality. Kruskal–Wallis tests with a Dunn’s post hoc were carried out for assessment of the age of tissue for each labeling category. Data are presented as mean ± standard error of the mean (SEM) or median ± quartiles, as indicated in the figure legends. Statistical significance was set at a *p* value < 0.05. 

## 3. Results

### 3.1. Central Fluid Percussion Injury Induces Significant Diffuse Axonal Injury in the Thalamus and the Corpus Callosum That Correlate with Microglial Activation

In our previous study [51], we reported TAI in various brain regions, including the thalamus and corpus callosum. We focused our previous assessments for that study on the thalamus, finding a significant increase in the degree of axonal injury within the thalamic domain (t_18_ = 4.46, *p* = 1.4 × 10^−5^) as well as a significant increase in the microglial activation index (t_4_ = 0.51, *p* = 0.004) [51]. For this study, we extended our evaluation of the degree of axonal injury and microglia activation to the corpus callosum, another brain region sustaining consistent TAI at 6 h following a central fluid percussion injury (CFPI) in our micro pig model [51]. We found that the degree of TAI, indicated as amyloid precursor protein (APP) accumulations at the proximal axonal swelling, within the corpus callosum was significantly higher at 6 h following CFPI (*n* = 18, 50.89 ± 19.26) compared with sham (*n* = 3, 0.64 ± 0.37; t_17_ = 1.44, *p* = 0.018; Figure 2A). Microglial activation, signified as an increase in the microglial activation index, was also significantly increased in the corpus callosum at 6 h following CFPI in our micro pig model (sham = 1.17 ± 0.13 CFPI = 3.96 ± 0.15; t_9_ = 2.51, *p* = 1.57 × 10^−7^; Figure 2B), suggesting that the corpus callosum also sustains significant TAI and microglial activation within 6 h of CFPI.

### 3.2. Central Fluid Percussion Injury Generates Scalable Diffuse Pathology That Does Not Significantly Impact Systemic Physiology

To investigate the potential of the micro pig cFPI model to scale the degree of pathology while maintaining a diffuse brain injury in which no cortical contusion was generated, we correlated the degree of TAI and microglial activation index within both the thalamus and the corpus callosum to the atmospheric pressure (ATM) induced in each individual animal generated in our 2015 study [51] (Figure 3). As the male micro pigs used in this study varied in terms of weight from 14.9 kg to 29.8 kg (Appendix A), we first normalized the ATM pressure to body weight. This normalized ATM pressure (ATM/kg) was significantly correlated with the raw ATM pressure transduced through the fluid percussion device (Rho = 0.96 *p* < 0.0001, Figure 3A), while allowing for specific weight-based refinement of the injury intensity. 

The body-weight-normalized ATM pressure positively correlated with TAI within both the thalamus (Spearman Rho = 0.45 *p* = 0.04) and the corpus callosum (Spearman Rho = 0.60 *p* = 0.004; Figure 3B). The degree of axonal injury within the corpus callosum also significantly correlated with the degree of TAI found in the thalamus (Spearman Rho = 0.71, *p* = 0.0003; Figure 3B), indicating that both regions scale with the intensity of the fluid pressure pulse. As previously published, thalamic TAI significantly correlates with the microglia activation index within the thalamus (Spearman Rho = 0.76 *p* = <0.0001) [51], indicating an interplay between microglial activation and TAI in this region. Microglial activation within the corpus callosum also significantly and positively correlated with TAI within the region (Spearman Rho = 0.64 *p* = 0.0018), indicating that this association is not thalamus-specific. 

The microglial activation index within the thalamus was found to be positively correlated with the microglial activation index within the corpus callosum (Spearman Rho = 0.54 *p* = 0.019; Figure 3B), suggesting that microglial activation might also scale with injury intensity in these two regions. While the microglial activation index within the thalamus was not correlated with the weight-normalized fluid pressure pulse (Spearman Rho = 0.29 *p* = 0.20), there was a significant positive correlation between ATM/kg and the microglial activation index within the corpus callosum (Spearman Rho = 0.68 *p* = 0.0008; Figure 3B). 

Post-injury blood gases were also assessed to investigate the potential that the intensity of cFPI alters systemic physiology. As we previously published, neither the CFPI nor sham groups had blood gases that were outside of the normal range (Appendix A). None of the individual animal’s blood measurements, including Oxygen saturation (O_2_%), partial pressure of O_2_ (PaO_2_), partial pressure of CO_2_ (PaCO_2_), mean arterial blood pressure (MABP), blood pH, hematocrit (Hct), bicarbonate in the blood (HCO_3_), or Hemoglobin (Hb), were correlated with weight-normalized pressure induction (Figure 4). This indicates that, while the diffuse pathology is correlated with the injury intensity within mild ranges, the overall systemic physiology reflected in the blood gases are not altered.

We did find significant positive correlations between diffuse pathology and subtle blood gas changes. The degree of TAI within the corpus callosum correlated with PaO_2_ (Spearman Rho = 0.51 *p* = 0.018), HCO_3_ (Spearman Rho = 0.47 *p* = 0.034), and blood pH (Spearman Rho = 0.46 *p* = 0.037). Thalamic TAI only correlated with blood pH (Spearman Rho = 0.44 *p* = 0.047), and thalamic microglia activation correlated with HCO_3_ (Spearman Rho = 0.44 *p* = 0.045). However, the lack of correlation between injury intensity and blood gases suggests that blood gas readouts are not a good metric of injury intensity in this model. 

### 3.3. Fixed Pig Tissue Maintains the Capacity for Immunohistochemical Labeling for Years after Refrigerated Storage

Fixed thalamic micropig tissue following long-term refrigerated storage was immunolabeled for astrocytes using GFAP and microglia, using Iba-1 to determine the tissue’s viability for use in immunofluorescent microscopy. Tissue harvested within 2 months of labeling was used as a positive control (Figure 5A). The degree of labeling for aged tissue was distinguished by the visibility of cell soma and processes, as well as the density of labeling within the images that were assessed. Samples were categorized as having a high degree of labeling (densely labeled cell soma and processes), a medium degree of labeling (sporadically labeled cell soma or processes), or a low/no-degree of labeling (non-labeled cell soma and processes; Figure 5B–D). Most of the cases assessed had a high-degree of labeling for both GFAP (58%) and Iba-1 (47%; Figure 5E). Medium-degree labelling was more prevalent in Iba-1 labelling (32%) versus GFAP (21%; Figure 5E). Only 21% of tissue did not label for GFAP or Iba-1, indicating that this tissue maintains some degree of immunofluorescent labeling capacity. 

As there was a difference in the efficacy of immunolabeling for GFAP and Iba-1, we further assessed the potential that the degree of immunolabeling was associated with the age of the tissue for each marker. The age of tissue did not significantly change when stratified by labeling degree for GFAP (X^2^(2) = 2.10, *p* = 0.35; Figure 5F). In contrast, the age of the tissue was significantly different when stratified by the degree of labeling for Iba-1 (X^2^(2) = 7.55, *p* = 0.023; Figure 5G). Specifically, the age of tissue with a high-degree of Iba-1 labeling was lower (94.8 ± 1.03 months) than tissue with a medium degree of Iba-1 labeling (101.5 ± 2.1 months; *p* = 0.011), and trended toward being lower than tissue with low/no Iba-1 labeling (102.3 ± 4.9; *p* = 0.065), suggesting that the age of the tissue impacts the labeling efficacy of Iba-1. 

## 4. Discussion

Mild TBI is associated with diffuse pathological progressions, including, but not limited to, TAI and neuroinflammation, which are associated with negative outcomes and morbidities in the human population [59,60,61,62]. Although classified as mild clinically based on their Glasgow comma score, mild TBI injuries occur on a spectrum, which is still not fully understood, but involves changes in the degree of diffuse pathology. As diffuse axonal injury as a clinical diagnosis can only be confirmed using histological assessment in postmortem tissue [17,18,63], detailed investigations into the spectrum of mild injury requires animal modeling. The cytoarchitecture, inflammatory system, and metabolism of pigs are highly consistent with humans [44,47,64]; therefore, we utilized a pig model to investigate the spectrum of diffuse injury produced by a CFPI model of mild TBI. 

In our initial study, we found significant axonal injury and microglia activation within the thalamic domain, which is consistent with findings in the human population [20,51,65,66,67]. We had noted TAI within the corpus callosum in that study as well; however, we had not previously quantitatively assessed the burden of TAI within the corpus callosum. In this study we found that our model of CFPI in the micro pig produced significant TAI and microglial activation within the corpus callosum (Figure 2). This is consistent with what is seen in the human population, with microscopic damage to the corpus callosum being one of the primary indicators of grade 1 diffuse axonal injury in the Adams diffuse axonal injury classification [68]. 

The burden of axonal injury within the corpus callosum and thalamus were significantly correlated with one another (Figure 3), demonstrating that in the pig CFPI model of mild TBI, both grey and white matter regions are impacted to a similar degree, allowing versatility for pathological assessments in this model. Axonal injury within both regions also demonstrated significant positive correlations to microglia activation following CFPI, indicating that microglia activation likely maps to areas of axonal injury. While we previously showed this correlation between TAI and microglia activation to be due to microglia process convergence onto injured axonal swellings in the thalamus in the pig following CFPI [50,51], further studies are needed to investigate the physical relationship between TAI and microglia activation in the corpus callosum. 

Importantly, there was a significant correlation between the intensity of the fluid pulse and the degree of TAI within both the thalamus and corpus callosum, showing that this pig model of mild TBI allows assessment of subtle changes in both grey and white matter areas along the spectrum of mild TBI. This finding is consistent with clinical investigations in which the degree of diffuse injury has been associated with severity of injury and progressive morbidity following TBI in experimental and clinical studies [8,9,59]. 

While the degree of TAI in both the corpus callosum and thalamus significantly correlated with fluid pulse intensity, only corpus callosal microglia activation was significantly correlated with the pressure pulse (Figure 3). As the microglia activation indices within the thalamus and corpus callosum were found to be significantly correlated, the lack of significance of thalamic microglia activation with injury intensity is likely an artifact of the assessment strategy. The assessment within the corpus callosum involved twice as many sections for analysis for each animal since the corpus callosum occupies a much smaller area than the thalamus. The lower degree of sampling within the thalamus of each animal could have introduced a higher degree of inter-animal variability. This difference between the correlation of the corpus callosum and thalamic microglia activation index with injury intensity could also be due to the more homogenous fiber orientations found within the corpus callosum compared to the thalamus which has a larger overall area and more heterogeneous axonal orientations. 

Interestingly, while the injury intensity was not significantly correlated with blood gas readouts and all blood gas readouts remained within normal ranges, diffuse pathology was found to be correlated with some blood gas readouts. Specifically, the degree of TAI within the corpus callosum significantly correlated with blood gas readouts, including paO2, pH, and HCO3, and the degree of thalamic TAI correlated with blood pH. However, this might not indicate a specific relationship between the brain pathology and subtle blood gas changes. Axonal injury and these subtle blood gas changes could both be symptoms of mild TBI that occur together but do not necessarily drive one another. Rather, the changes in blood gases could indicate subtle changes in systemic organs following a mild TBI. Studies have recently shown that brain injury can induce changes within systemic organs systems, including both the kidney and lung, both regions involved in regulating the HCO3, pH, and PaO2 of the blood. Kidney and lung pathology has been shown to occur in rodent models of mild TBI [69,70,71,72,73]. Retrospective clinical studies also found that either acute kidney injury [74] or acute lung injury [75] were associated with worse outcomes in TBI patients, indicating that systemic organ changes could be occurring in the human population as well. These are intriguing possibilities that require further investigation; however, due to the blood gas readouts in the current study all being within normal ranges, and the lack of correlation between any blood gas readout and the pressure pulse (Figure 4), it does not appear that CFPI substantially changes the blood gas readouts. Therefore, it is likely that whatever potential changes might be occurring systemically, they are not extreme enough to break homeostasis.

In terms of the ability to label tissue years following generation, most of the tissue stored in long-term (7.5–9.6 years) refrigeration expressed a high degree of immunolabeling of both astrocytic GFAP and microglial IBA1 markers. These data suggests that the efficacy of GFAP labeling is not altered as the age of tissue increases. Iba-1, on the other hand, demonstrates more medium and low/no degree of labeling in older tissues, suggesting that the specific antibody is less effective in older tissues. The Wako Iba-1 antibody has been used in various species to identify microglia with a high degree of efficacy [76,77,78,79]. The foundational studies using this Iba-1antibody all appeared to use transcardial perfusion with 4% paraformaldehyde followed by post-fixation [77,78,79]. Our prior published studies with the same tissue indicates that the Iba-1 antibody did not suffer decreased labeling in newly generated tissues as it did with tissues from long-term storage. This may suggest that the Wako IBA-1 antibody loses efficacy in older tissues. 

## 5. Conclusions

Overall, the current study shows that a mild model of central fluid percussion injury can produce scalable diffuse injury within multiple brain regions without significantly altering blood gases. Additionally, we found that the majority of tissue fixed with 4% paraformaldehyde/0.2% glutaraldehyde and stored in Millonig’s buffer under refrigeration conditions is usable for immunofluorescent analysis up to 8 years following generation. 

## Figures and Tables

**Figure 1 biomedicines-11-01682-f001:**
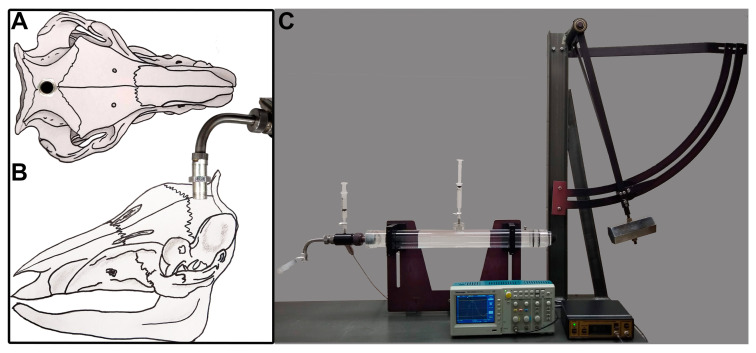
Adult male micro pigs were subjected to either sham injury or a mild central fluid percussion injury (CFPI). Representative drawings of (**A**) dorsal and (**B**) lateral view of the pig skull with the injury hub screwed into the skull along the sagittal suture. (**C**) Picture of the fluid percussion device used to induce CFPI in the micro pig. The injury hub was screwed into a 16 mm craniectomy placed 14 mm anterior to the nuchal crest. The hub was filled with sterile saline then connected to the fluid percussion device’s L-shaped adaptor. Following connection to the device, the hammer was released, producing a fluid pressure pulse that transduced through the dura to the cerebrospinal fluid.

**Figure 2 biomedicines-11-01682-f002:**
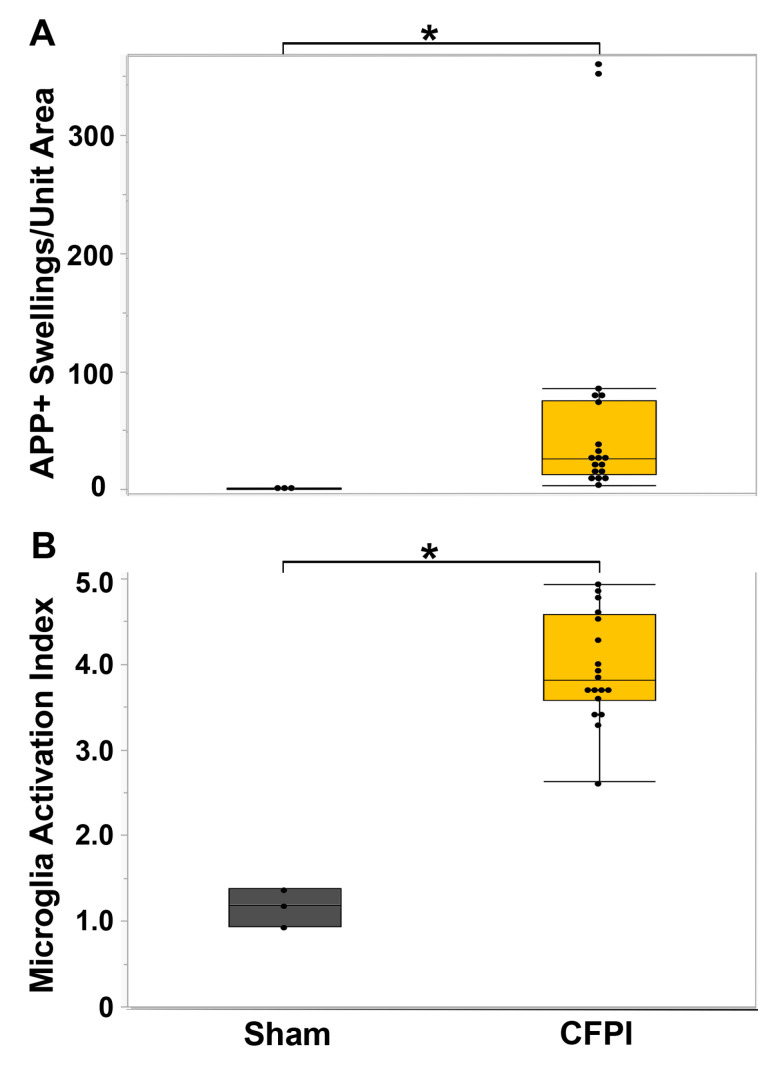
Central fluid percussion injury results in significant axonal injury and microglial activation within the corpus callosum. Box plots of (**A**) APP+ Axonal swellings indicative of diffuse traumatic axonal injury and (**B**) microglial activation index at 6 h following sham (*n* = 3, grey boxes) or cFPI (*n* = 18, yellow boxes). Each black circle indicates an individual animal. Graphs depict the median ± quartile. * *p* < 0.05.

**Figure 3 biomedicines-11-01682-f003:**
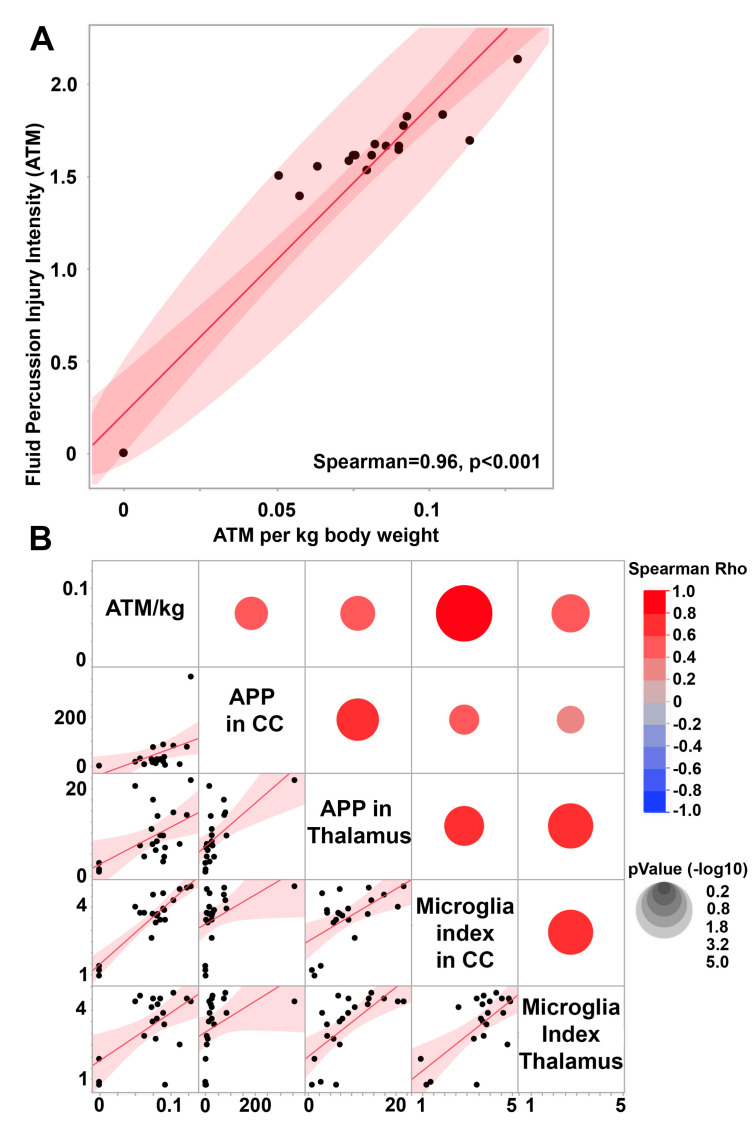
Weight-normalized central fluid percussion pressure pulse positively correlates with diffuse traumatic axonal injury and microglia activation in the thalamus and corpus callosum. (**A**) Scatter plot of total fluid percussion ATM pressure and the ATM pressure normalized to kg weight. (**B**) Correlation matrix depicting the Spearman Rho correlation for the weight-normalized injury intensity (ATM/kg) with the degree of axonal injury in the corpus callosum (APP in CC) or thalamus (APP in Thalamus) as well as the degree of microglia activation in the corpus callosum (Microglia index in CC) or thalamus (Microglia index in Thalamus). Each pair of assessment values on the *y* and *x* axis is colored with red, indicating a positive correlation, or blue, indicating a negative correlation, with the correlation strength demonstrated by the circle’s size, with larger circles representing lower *p* values. The lower part of the correlation matrix shows the correlation curves for each x and y comparison with each black dot representing data from an individual animal. Note that degree of axonal injury and microglial index values show positive correlations with ATM/kg, suggesting scalability between the pressure pulse inflicted and the degree of injury as well as the level of neuroinflammation in both regions.

**Figure 4 biomedicines-11-01682-f004:**
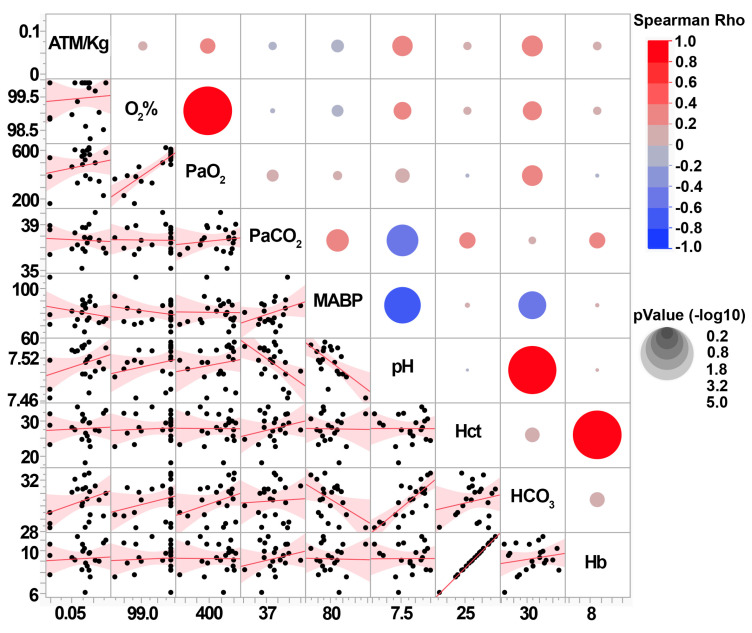
Blood gas readings do not correlate with cFPI injury intensity (ATM/kg) in micro pigs. Correlation matrices depicting the Spearman Rho correlation for the weight-normalized injury intensity (ATM/kg) with the various blood gas measurements and the mean arterial blood pressure (MABP) 6 h post-CFPI. Each pair of assessment values on the *y* and *x* axis is colored with red indicating a positive correlation, or blue indicating a negative correlation, with the correlation strength demonstrated by the circle’s size. The lower part of the correlation matrix shows the correlation curves for each x and y comparison, with each black dot representing data from an individual animal.

**Figure 5 biomedicines-11-01682-f005:**
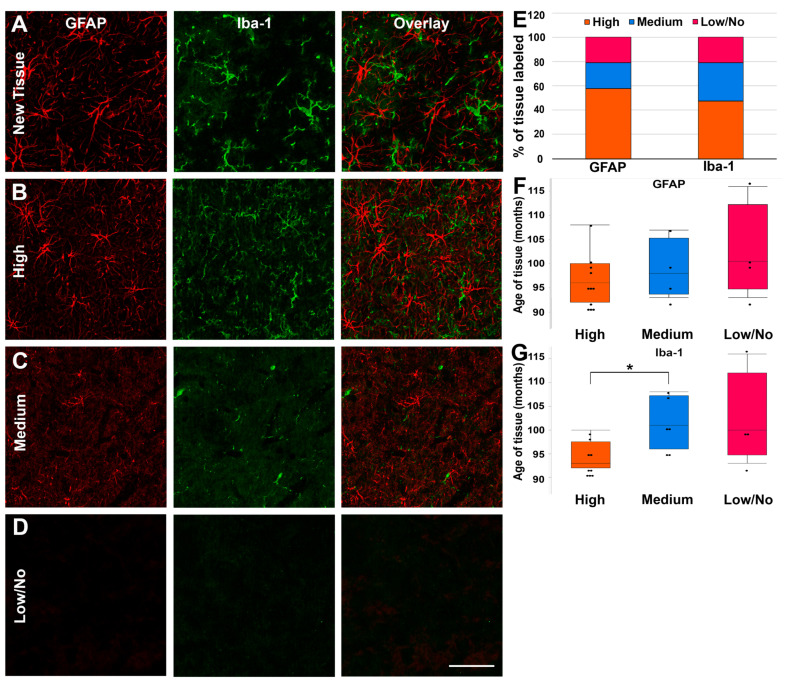
The majority of long-term stored labeled tissues exhibited a high degree of labeling of both GFAP and Iba-1. (**A**–**D**) Representative epifluorescent micrographs show astrocytes labeled with GFAP (first column; red) and microglia with Iba-1 (second column; green). The third column shows an overlay of GFAP and Iba-1. Each row shows representative images of (**A**) freshly harvested new tissue as a positive labeling control, and tissued stored for >7 years with (**B**) high-degree, (**C**) medium-degree, and (**D**) low/no-degree labeling of GFAP and IBA1. Scale bar: 50 μm. (**E**) Stacked bar graphs show the proportional number of tissues that were categorized into high-degree, medium-degree, and low/no-degree labeling for both GFAP and IBA1. (**F**,**G**) Box plots depicting the interquartile range and median age of tissue of each animal labeled with (**F**) GFAP or (**G**) IBA1 are categorized by degree of labeling. Note that the mean ages of GFAP-labeled tissue are less variable between labeling degrees, while the mean age of IBA1 tissue increases significantly from high-degree to medium-degree-labeled tissue. Graphs depict the median ± quartile. Black dots denote individual tissue samples. * *p* < 0.05.

## Data Availability

Full data for each animals can be found in Appendix A.

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
