# Peer review of "The Central Fluid Percussion Brain Injury in a Gyrencephalic Pig Brain: Scalable Diffuse Injury and Tissue Viability for Glial Cell Immunolabeling following Long-Term Refrigerated Storage"

_biomedicines, 2023, doi:10.3390/biomedicines11061682_

Round 1

Reviewer 1 Report

The authors have submitted a well written manuscript detailing fluid percussion experiments in pigs. Minor revisions are requested.

1.     Figure 1. Statistical significance is indicated. However the * on the graph is not present. It would be easier to identify the results if it were present (as many readers just review the graphs and don't read all the text).

2.     Figure 5. The fluroescence labeling is very dim and hard to evaluate (even in the high labeling group). Can freshly harvested tissue be shown for comparison? What about differences in the morphology? Does the labeling increase if the antibody concentration is increased?

3.     Line 454-455. Incomplete sentence.

Author Response

We thank the reviewers for their time and comments. We have edited the manuscript according to the suggestions using track changes.

  1. Figure 1. Statistical significance is indicated. However the * on the graph is not present. It would be easier to identify the results if it were present (as many readers just review the graphs and don't read all the text).

We have now included * on the graphs in Figure 2 to indicate significant differences between the groups.

  1. Figure 5. The fluorescence labeling is very dim and hard to evaluate (even in the high labeling group). Can freshly harvested tissue be shown for comparison? What about differences in the morphology? Does the labeling increase if the antibody concentration is increased?

We have now included freshly harvested tissue as a positive control for the labeling in Figure 5. We have also enhanced the brightness of all micrographs shown in Figure 5.

  1. Line 454-455. Incomplete sentence.

We have now completed that sentence: “This may suggest that the Wako IBA-1 antibody loses efficacy in older tissues”

Reviewer 2 Report

The aim of this study was to evaluate the use of a central fluid percussion injury in a micro pig model for investigation of diffuse axon injury.

The Introduction is very clearly written and all references are appropriate and relevant. The reason for the micro pig model is also clearly explained.

The methodology is very clearly explained and Figure 1 is helpful for the reader to understand the tool used for CFPI. The statistical analyses used are also appropriate and clearly explained.

Results are also clearly explained.

Line 280: Change "compared to" to "compared with"

Figure 2: Sham and CFPI can be removed from Figure 2A. Add a space between "Figure" and "2"

The Discussion section is clearly written and explains the results relative to the current body of knowledge.

Lines 453-454: References needed.

Author Response

We thank the reviewers for their time and comments. We have edited the manuscript according to the suggestions using track changes.

1) Line 280: Change "compared to" to "compared with"

This change has been made

2) Figure 2: Sham and CFPI can be removed from Figure 2A. Add a space between "Figure" and "2"

We have removed “Sham” and “CFPI” from Figure 2A and added the space between Figure and 2.

3 )Lines 453-454: References needed.

We have added the reference to Figure 4 for the statement “While these are intriguing possibilities that require further investigation; due to the blood gas readouts in the current study all being within normal ranges and the lack of correlation between any blood gas readout and the pressure pulse (figure 4), it does not appear that CFPI substantially changes the blood gas readouts.”